# Reward Collapse in Aligning Large Language Models

## Abstract

The extraordinary capabilities of large language models (LLMs) such as ChatGPT and GPT-4 are in part unleashed by aligning them with reward models that are trained on human preferences represented as rankings of responses to prompts. In this paper, we document the phenomenon of *reward collapse*, an empirical observation where the prevailing ranking-based approach results in an *identical* reward distribution for diverse prompts during the terminal phase of training. This outcome is undesirable as open-ended prompts like "write a short story about your best friend" should yield a continuous range of rewards for their completions, while specific prompts like "what is the capital city of New Zealand" should generate either high or low rewards. Our theoretical investigation reveals that reward collapse is primarily due to the insufficiency of the ranking-based objective function to incorporate prompt-related information during optimization. This insight allows us to derive closed-form expressions for the reward distribution associated with a set of utility functions in an asymptotic setting. To overcome reward collapse, we introduce a prompt-aware optimization scheme that provably admits a prompt-dependent reward distribution within the interpolating regime. Our experimental results suggest that our proposed prompt-aware utility functions significantly alleviate reward collapse during the training of reward models.

## 1 Introduction

A cornerstone of the recent remarkable advancements in the capabilities of large language models (LLMs) like ChatGPT and GPT-4 is the integration of human feedback (Ouyang et al. (2022); OpenAI (2023)). The approach to leveraging human feedback often begins with the training of a reward model that encapsulates human preferences, values, and ethical considerations (Christiano et al. (2017); Ibarz et al. (2018); Bahdanau et al. (2018); Ziegler et al. (2019); Ganguli et al. (2022)). This is followed by the fine-tuning of the LLMs using reinforcement learning, guided by the reward model. This process, often referred to as reinforcement learning from human feedback (RLHF), has proven effective in aligning LLMs with human intent, substantially enriching the quality of human interaction.

However, developing an effective reward model based on human preferences is challenging (Bai et al. (2022b); Liu et al. (2023); Sun et al. (2023)). A notable difficulty arises when a human labeler struggles to give a quantitative score to a response/completion for a specific prompt. Instead, it is much easier for humans to make pairwise comparisons between completions in terms of their quality, which is indeed employed in the development of InstructGPT (Ouyang et al. (2022)). Explicitly, a human labeler is presented with several completions generated by the LLMs for the same prompt and arranges the responses from the highest to lowest perceived quality.[1] A neural network is then trained to obtain a reward model that assigns rewards to the responses in an attempt to align as closely as possible with human preferences in the form of *rankings*.

Despite some benefits, such as eliminating calibration issues, rankings fall short in reflecting the varied reward distributions of different prompts. This is due to the fact that ranking one completion higher than another does not indicate how *much* superior the former is compared to the latter. This concern is especially pertinent in RLHF as some prompts are open-ended or, in other words, are dependent on the users' backgrounds, allowing the reward distribution to span a continuous range. Conversely, some prompts are closed-ended,

---

[1]In slightly more detail, Ouyang et al. (2022) required human labelers to utilize a drag-and-drop interface to construct *consistent* rankings from pairwise comparisons.

resulting in a response that should be either highly or lowly scored, thus generating a roughly two-point mass distribution for the reward distribution. Instances of the first type of prompts include *write a short story about how AI will look like in 100 years* and *what is the best cuisine in the world*, while examples of the second type are *prove the Pythagorean theorem* and *is chicken a dinosaur*. An ideal reward model would assign a reward of either low or high to close-ended prompts, ensuring that the completion accurately aligns with the correct direction. Conversely, for open-ended prompts, the reward should avoid being either low or high to encourage diverse responses. If the reward model cannot distinguish between open-ended and close-ended prompts, it fails to assist language models in determining uncertainty when providing completions, whether with high variability or low variability (Padmakumar & He (2023)). As a result, the reward model may struggle to aid LLMs in accurately calibrating uncertainty without accounting for the nuances of different prompts. [2]

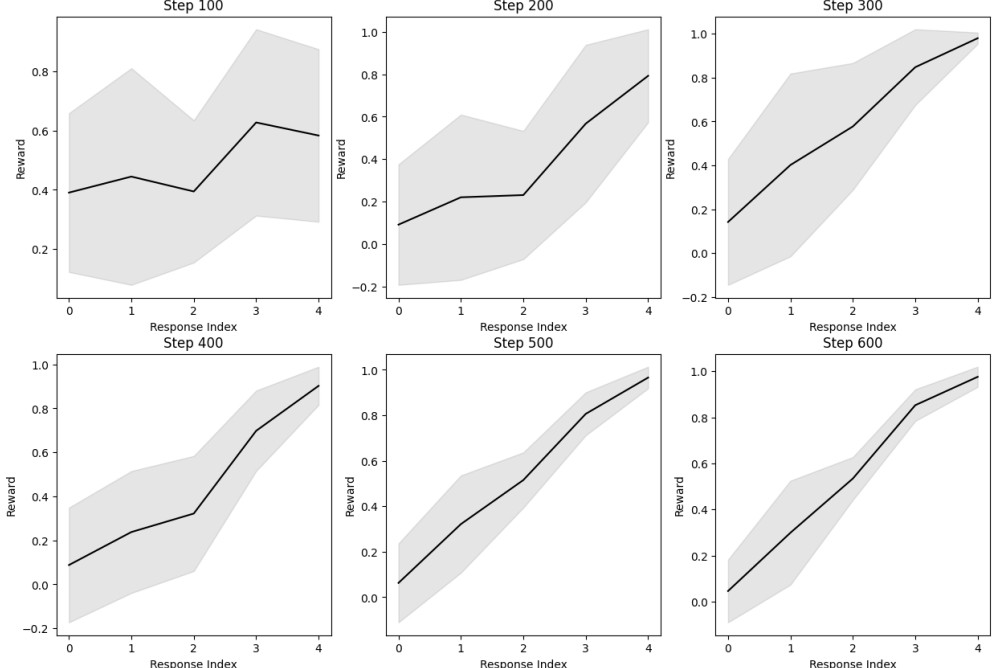

Figure 1: Reward distribution of the five responses throughout the training process. The $x$-axis represents the response index, sorted by reward from smallest to largest. The solid curve illustrates the mean across several prompts, while the shadowed area represents the standard deviation. A clear observation from the figure reveals the progressive convergence of the distribution towards a single value, thereby evidencing the reward collapse phenomenon. Experiment details are elaborated in Section 5.

As our first main contribution, this paper documents a surprising phenomenon through a series of experiments, demonstrating that training a reward model on preference rankings could result in the *same* reward distribution regardless of the prompts. We call this phenomenon *reward collapse*, which occurs during the terminal phase of training Papyan et al. (2020). Intriguingly, our theoretical analysis first predicted this phenomenon prior to its experimental confirmation. Indeed, we show that the collapsed reward distribution can be numerically deduced from a simple optimization program or, even simpler, admits a closed-form expression. As demonstrated in Figure 1, our prediction of reward collapse is in agreement with the empirical results.

Reward collapse is clearly undesirable as it overlooks the subtle differences among various prompts, potentially leading to the miscalibration of human preference during the training of LLMs via reinforcement learning with the reward model. A rudimentary strategy to bypass this issue is to early stop the training of

---

[2]For instance, we suspect that this is partly accountable for the poor calibration of GPT-4 after RLHF (see page 12 of OpenAI (2023)) and mode collapse (Casper et al. (2023a;b)).

the reward model Ouyang et al. (2022), which, however, fails to address the fundamental limitation of using a single utility function across all prompts.

In our second main contribution, we introduce a principled approach to alleviating reward collapse, leveraging insights derived from the same optimization program that was instrumental in predicting this phenomenon. In essence, we propose to use distinct utility functions depending on prompts in training the reward model, such that the resulting reward distribution can be either widely dispersed or tightly concentrated, contingent on whether the prompt is open-ended or closed-ended. A notable advantage of this prompt-aware strategy is that our analysis is analytical, enabling full control over the shape of the reward distribution as required. Our experiments show that reward collapse can be substantially mitigated using this prompt-aware methodology.

## 2 What Is Reward Collapse

### 2.1 Reward modeling

We use $x$ and $y$ to denote prompts and completions. And we use $R(x, y)$ to denote a reward model. In this paper, we assume $R(x, y) \in [0, 1]$. For a given prompt and $n$ completions that are i.i.d. draws from an LLM, a human labeler ranks the $n$ responses from the most preferred to the least preferred, and the ranking is denoted as $\pi_x$. The dataset is given by[3]

$$\mathcal{D} = \{(x, y_1, \cdots, y_n) : \ x \text{ is a prompt,}$$

$$y_1, \cdots, y_n \text{ are its completions from the most prefered to the least prefered}\}$$

The reward model is expected to score each completion that is consistent with the human-provided ranking $\pi_x$ as much as possible. To this end, we train a neural network that maximizes the following overall utility:

$$\sum_{(x, y_1, \cdots, y_n) \in \mathcal{D}} \sum_{1 \le i < j \le n} U\left(R_\theta(x, y_i) - R_\theta(x, y_j)\right), \tag{1}$$

where $U$ is an (increasing) utility function, $\theta$ is the weights of the reward neural network. Typically, $U$ is set to $U(z) = \log \mathtt{sigmoid}(cz) \equiv \log \frac{e^{cz}}{e^{cz}+1}$, which is an increasing concave function (Ouyang et al. (2022); Rafailov et al. (2024)). While maximizing Eq. 1, the reward model learns to not only align with the human-provided ranking but also distinguish the rewards as much as possible.

### 2.2 Reward collapse

To illustrate what is reward collapse, we start with the overall utility (1). Let

$$S(r_1, \cdots, r_n) = \sum_{1 \le i < j \le n} U\left(r_i - r_j\right) \tag{2}$$

then Eq.1 can be further written as

$$\sum_{(x, y_1, \cdots, y_n) \in \mathcal{D}} S(R_\theta(x, y_1), \cdots, R_\theta(x, y_n)).$$

Consequently, if the maximum of $S(r_1, \cdots, r_n)$ is $M$, overall utility (1) is upper bound by $|\mathcal{D}|M$. Furthermore, if $\hat{r}_1, \cdots, \hat{r}_n$ is the unique maximizer of $S(r_1, \cdots, r_n)$ with $r_1 \ge \cdots \ge r_n$, then overall utility can reach $|\mathcal{D}|M$ if and only if

$$R_\theta(x, y_i) = \hat{r}_i, i = 1, \cdots, n. \tag{3}$$

In fact, for any reward model that sufficiently optimize the overall utility, the reward $R_\theta(x, y_i)$ is close to $\hat{r}_i$ for all prompts. We call this phenomenon *Reward Collapse*. (equation 3). Formally, we have the following theorem:

---

[3]Here, we assume that each prompt has the same number of completions. However, our theory can be readily generalized to cases where each prompt has a different number of completions.

**Theorem 1** (Reward collapse)**.** *Assume $U$ is strongly concave with parameter $\mu > 0$ and strictly increasing, then $S$ defined in (2) has some maximum $M$ obtained uniquely at $\hat{r}_1, \cdots, \hat{r}_n$. For any neural network parameterized by $\theta$, such that*

$$\sum_{(x,y_1,\cdots,y_n)\in\mathcal{D}} \sum_{1\leq i<j\leq n} U\left(R_\theta(x,y_i) - R_\theta(x,y_j)\right) \geq |\mathcal{D}|M - \frac{\mu n \epsilon^2}{2},$$

*we have*

$$\max_i |R_\theta(x,y_i) - \hat{r}_i - c(x)| \leq \epsilon$$

*for all $(x, y_1, \cdots, y_n) \in \mathcal{D}$ and some constant $c(x)$ depending on $x$.*

That is, the empirical distribution of the rewards is approximately independent of the prompt itself in the interpolating regime, thereby leading to reward collapse. The proof of this theorem can be found in Appendix B

To further illustrate which neural network maximizes overall utility, consider the case where reward function is parameterized as $R_\theta(x,y) = \sigma(\langle\theta, \phi(x,y)\rangle)$, where $\sigma$ is the sigmoid function, $\theta \in \mathbb{R}^d$ represents the weights, and $\phi(x,y) : \mathcal{X} \times \mathcal{Y} \to \mathbb{R}^d$ is a known and fixed feature function. Such a reward parameterization is usually derived by removing the last layer of the pre-trained model. A similar parametrization is also used in Zhu et al. (2023). Note that we include a sigmoid function to ensure that the reward is in $[0, 1]$. Then if $S$ defined in (2) attains its maximum $M$ uniquely at $\hat{r}_1, \cdots, \hat{r}_n$ and $d \geq |\mathcal{D}|n$, there exist a $\theta^*$, such that

$$R_{\theta^*}(x, y_i) = \hat{r}_i, i = 1, \cdots, n.$$

Consequently, when training an over-parameterized neural network maximizing the overall utility, it is likely to observe reward collapse. We validate this theoretical result through experiments on large language models, as detailed in Section 5.

In practice, reward collapse is not what we want to observe in the reward model. Consider the following case where two prompts are given: one open-ended, such as "write a short story about your best friend," and one close-ended, such as "what is the capital city of New Zealand." We expect the rewards for different responses to the open-ended prompt to be continuously distributed within $[0, 1]$. However, for the close-ended prompt, the rewards for different responses should be either 0 or 1. The reward model needs to provide different reward distributions for different kinds of prompts.

## 3 Prompt-aware optimization

To avoid having the same reward distribution, one simple strategy is early stopping. While reward collapse can be avoided via early stopping, early stopping might make the model neglect other important features. A more principled approach is to change the objective. Our proposal is to let the utility function $U$ now depend on the prompt. That is, now we consider training a neural network that maximizes

$$\sum_{(x,y_1,\cdots,y_n)\in\mathcal{D}} \sum_{1\leq i<j\leq n} U_x\left(R_\theta(x,y_i) - R_\theta(x,y_j)\right), \tag{4}$$

where $U_x$ is a utility function that depends on the prompt $x$. Note that this reward modeling approach is similar to traditional reward modeling in that it also aims to maximize the difference in rewards while incorporating ranking information. However, the key difference here is that we allow for different utility functions for different prompts $x$, making the reward model more sensitive to variations in prompts and, hence, more accurately indicating the effect of different prompts.

In general, the choice of $U_x$ should reflect the open-endedness of the prompt $x$. Given the high flexibility in choosing $U_x$, it is generally recommended to let the practitioners choose these functions to meet their needs. Nonetheless, below we introduce a family of such functions.

For a strictly increasing utility function $U$, it can be easily demonstrated that the maximum can only be attained when $r_1 \geq \cdots \geq r_n$ (see Lemma C.1 in the Appendix). As a result, we consider the problem

$$\max_{0 \leq r_n \leq \ldots \leq r_1 \leq 1} \sum_{1 \leq i < j \leq n} U\left(r_i - r_j\right). \tag{5}$$

We use the term "reward distribution" to refer to the empirical distribution of solutions to (5).

**Class 1.** Let $U(z) = z^\gamma, z \in [0, 1]$ for some $0 < \gamma < 1$. This utility function encourages the reward to take values either near 0 or 1 as $\gamma$ tends to be large. Some plots showing the reward distribution is given in Figure 2(a) and 2(b).

**Class 2.** Let $U(z) = -z^\gamma, z \in (0, 1]$ for $0 < \gamma \leq 1$. We also define $U(0) = \infty$ for $0 \leq \gamma \leq 1$. In this case, the reward distribution of Eq. 5 becomes more even as $\gamma$ increases from 0 to 1. Some plots are shown in Figure 2(c) and 2(d).

**Class 3.** Let $U(z) = \log \text{sigmoid}(z/\sigma), z \in [0, 1]$ for $\sigma > 0$. The reward distribution becomes more spread between 0 and 1 as $\sigma$ becomes smaller. Some plots are shown in Figure 2(e) and 2(f).

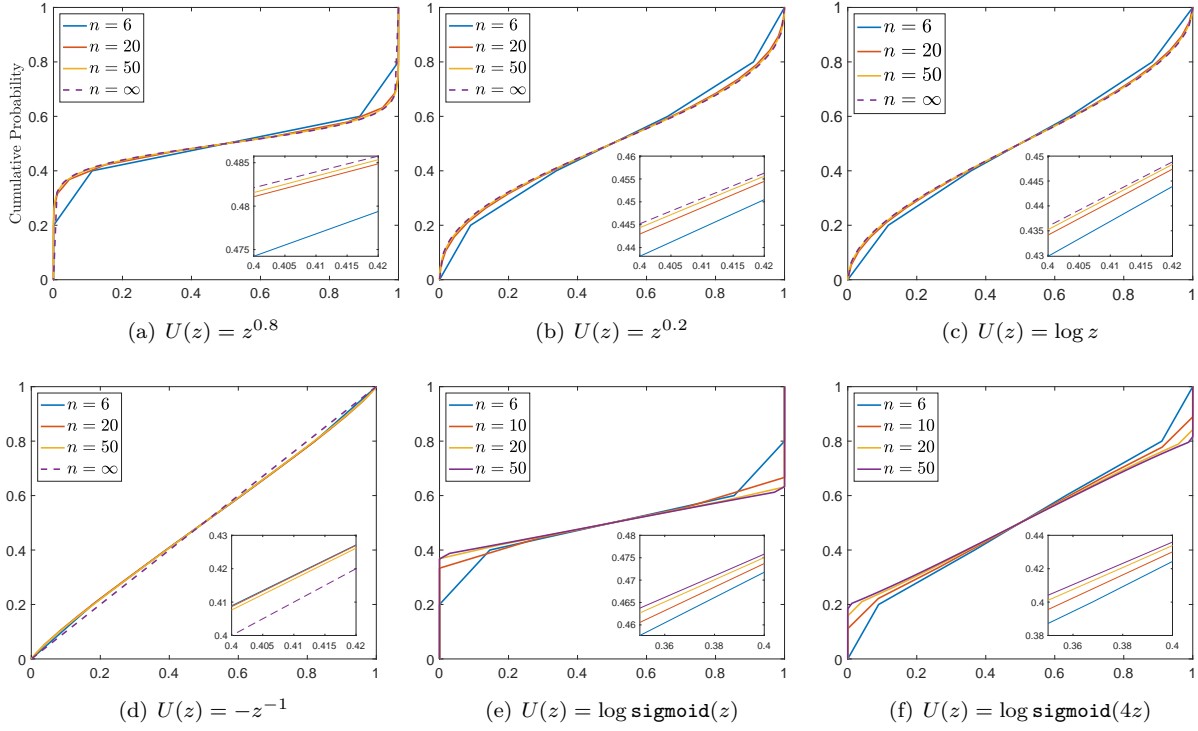

Figure 2: Empirical cumulative distribution function (e.c.d.f.) of rewards for different utility functions. As the number of responses $n$ increases, the e.c.d.f. converges to a limiting distribution.

## 3.1 Asymptotics

In general, we can explicitly evaluate the reward distribution for any $n$ by solving the optimization (5). Nevertheless, it is helpful to get a handle on the empirical distribution of the solution to this optimization program in the limit $n \to \infty$. The next result gives a closed-form expression of the reward distribution in the case of a large number of completions.

**Theorem 2.** Let $U(z) = z^\gamma, z \in [0, 1]$ for some $\gamma \in (0, 1)$. Then the reward distribution of (5) converges to Beta $\left(\frac{1-\gamma}{2}, \frac{1-\gamma}{2}\right)$ as $n \to \infty$, which has probability density $x^{-\frac{1+\gamma}{2}}(1-x)^{-\frac{1+\gamma}{2}}$ on $(0, 1)$.

**Theorem 3.** *For $U(z) = -z^{-\gamma}, z \in (0,1]$ for $0 < \gamma \leq 1$, the reward distribution of (5) converges in distribution to $\text{Beta}(\frac{1+\gamma}{2}, \frac{1+\gamma}{2})$. For $U(z) = \log z, z \in (0,1]$, the reward distribution of (5) converges in distribution to $\text{Beta}(\frac{1}{2}, \frac{1}{2})$*

The proof of Theorem 3 can be found in Martinez-Finkelshtein et al. (2004); Landkof & Landkof (1972). In the limit $\gamma \to 1$ in Theorem 3, the Beta distribution tends to $\text{Beta}(1,1)$, which is the uniform distribution on $[0,1]$. This is indeed an example of the one-dimensional Thomson problem (Bowick et al. (2002)), which asks the configuration of $n$ electrons constrained to a line that repel each other with a force given by Coulomb's law. This problem was first considered by Maxwell. Indeed, Martinez-Finkelshtein et al. (2004); Hardin et al. (2004); Amore & Jacobo (2019) prove that the reward distribution will converge to the uniform distribution for $U(z) = -z^{-\gamma}$ with $\gamma \geq 1$.

For the above two classes, the limiting distribution does not admit a probability mass. However, probability mass can emerge in the case of a scaled log-sigmoid function.

**Theorem 4.** *If $U$ is strictly increasing and concave, the derivative of the utility function satisfies $U'(0) < \infty, U'(1) > 0$, then the reward distribution of (5) converges in distribution to a probability measure $\mu^*$ that satisfies*

$$\mu^*(\{0\}) = \mu^*(\{1\}) \geq \frac{U'(1)}{U'(0)+U'(1)} > 0.$$

In general, the reward distribution can be characterized from a variational perspective. This gives the following theorem.

**Theorem 5.** *If $U$ is bounded, strongly concave, and increasing. There exists a probability measure $\mu^*$ such that the reward distribution of (5) converges in distribution to $\mu^*$, which is uniquely determined by the following two properties:*

*(a) $\mu^*$ maximizes*

$$\mathbb{E}_{X,X' \overset{iid}{\sim} \mu} U(|X - X'|)$$

*over all probability measures $\mu$ on $[0,1]$, and*

*(b) it is symmetric with respect to $\frac{1}{2}$ in the sense that, for any measurable set $A \in [0,1]$ and $1 - A = \{x : 1 - x \in A\}$, $\mu^*(A) = \mu^*(1 - A)$.*

### 3.2 Prompt-aware optimization based on open-endedness

Based on the asymptotic properties discussed, we propose a prompt-aware optimization approach that leverages the concept of open-endedness.

For a given prompt $x$, if it is close-ended (e.g., "What is the capital city of New Zealand?"), the reward for a response $R(x, y)$ should be either high or low, indicating a clear right or wrong answer. In such cases, we set $U_x(z) = z$, as its limiting reward distribution follows a Bernoulli distribution. Conversely, for an open-ended prompt (e.g., "Write a short story about your best friend"), the reward should span a continuous range, reflecting the diversity of possible responses. Here, we choose $U_x(z) = -z^{-1}$ to capture this variability. Mixed-type prompts, such as "What is the capital city of New Zealand? Tell me some interesting stories about it," require a response that addresses both factual accuracy and creative content. For these prompts, a natural choice is the log-sigmoid function, as its limiting distribution approximates a mixture of Bernoulli and uniform distributions, effectively balancing the different types of responses required.

## 4 Proofs

In this section, we will briefly present the proofs of results in Section 2. However, we will deviate from the previous order and start by proving Theorem 5. We also put the proof of Theorem 4 into Appendix D.3 due to the length constraint. Let

$$S(r_1, \cdots, r_n) := \sum_{1 \leq i < j \leq n} U(r_i - r_j) \text{ and } \hat{\mathbf{r}} \equiv (\hat{r}_1, \ldots, \hat{r}_n) := \arg \max_{0 \leq r_1, \cdots, r_n \leq 1} S(r_1, \cdots, r_n).$$

In addition, for any vector $(u_1, \cdots, u_n) \in \mathbb{R}^n$, we employ boldface notation $\mathbf{u}$ to represent the entire vector. THis allows us to write $S(\mathbf{r})$.

## 4.1 Proof of Theorem 5

First, when $U$ is concave and strictly increasing, $\hat{\mathbf{r}}$ exhibits the following properties:

**Lemma 4.1.** *If $U$ is strictly concave and strictly increasing, the function $S(\mathbf{r})$ is concave. Therefore, the optimization problem uniquely determines $\hat{\mathbf{r}}_n$. Additionally, the following properties hold: (1) $\hat{r}_1 \geq \cdots \geq \hat{r}_n$, and (2) $1 - \hat{r}_i = \hat{r}_{n-i+1}$ for any $1 \leq i \leq n$.*

The proof of Lemma 4.1 is straightforward and is provided in Appendix C.1. Upon further examination of the function $S(\mathbf{r})$, we discover that if $U$ is strongly concave with parameter $\mu > 0$, then $S$ also exhibits some kind of strongly concavity, except in the direction $(1, 1, \cdots, 1)$. This property is formulated in the following lemma.

**Lemma 4.2.** *If $U$ is strongly concave with parameter $\mu > 0$, and we consider another vector $\mathbf{u} = (u_1, \ldots, u_n)$, the following inequality holds:*

$$S(\mathbf{u}) - S(\hat{\mathbf{r}}) \leq -\frac{n\mu}{2} \| \operatorname{Proj}_{V_n}(\mathbf{u} - \hat{\mathbf{r}}) \|^2.$$

*Here, $V_n \subset \mathbb{R}^n$ is the subspace orthogonal to $(1, \cdots, 1)$, and $\| \cdot \|$ represents the Euclidean norm.*

The proof of this lemma can be found in Appendix C.2. Our next lemma quantifies the difference between two symmetric probability measures.

**Lemma 4.3.** *For two different symmetric probability measure $\mu_1$ and $\mu_2$ on $[0,1]$, let $r_i^{(j)} = \frac{1}{2} \inf\{t : \mu_j([0,t]) \geq \frac{n-i}{n-1}\} + \frac{1}{2} \sup\{t : \mu_j([0,t)) < \frac{n-i}{n-1}\}), i = 1, 2, \cdots, n; j = 1, 2$. Then there exists positive constant $c_0$ such that for all $n$,*

$$\| \operatorname{Proj}_{V_n}(\mathbf{r}^{(1)} - \mathbf{r}^{(2)}) \|_2^2 \geq c_0 n.$$

The proof of Lemma 4.3 is also provided in Appendix C.3. Now, we are ready to prove the uniqueness part of Theorem 5. Due to the length constraint, we will present it as a separate lemma and defer the proof to Appendix C.4. In short, we use Lemma 4.2 and 4.3 to demonstrate that for two distinct symmetric measures, their distance is sufficiently large such that at least one of them is not optimal.

**Lemma 4.4.** *If $\mu_1$ and $\mu_2$ are two symmetric probability measure which both maximize*

$$\mathbb{E}_{X,X' \overset{iid}{\sim} \mu} U(|X - X'|)$$

*over all probability measures $\mu$ on $[0,1]$. Then we have $\mu_1 = \mu_2$.*

Now we are ready to prove the convergence part of Theorem 5.

*Proof of Theorem 5.* Let $\hat{\mathbb{P}}_n := \frac{1}{n} \sum_{i=1}^n \delta_{\hat{r}_n}$ denote the empirical distribution of $\hat{\mathbf{r}}_n$. Note that $\{\hat{\mathbb{P}}_n\}$ are probability measures defined on $[0,1]$, so they are tight. By Prohorov's theorem, there exists a sub-sequence $\{k(n)\}_{n \geq 1}$ such that $\hat{\mathbb{P}}_{k(n)} \overset{d}{\to} \hat{\mu}$. Let $X_n, X_n' \overset{iid}{\sim} \hat{\mathbb{P}}_n$ and $\hat{X}, \hat{X}' \overset{iid}{\sim} \hat{\mu}$. By continuous mapping theorem, we also have $|X_n - X_n'| \overset{d}{\to} |\hat{X} - \hat{X}'|$. Moreover, because $U$ is bounded and continuous, Portmanteau theorem gives

$$\mathbb{E}_{X,X' \overset{iid}{\sim} \hat{\mathbb{P}}_{k(n)}} U(|X - X'|) \to \mathbb{E}_{X,X' \overset{iid}{\sim} \hat{\mu}} U(|X - X'|).$$

Let $\mu$ be another probability measure on $[0,1]$. Let $\hat{\mathbb{Q}}_n = \frac{1}{n} \sum_{i=1}^n \delta_{q_{n,i}}$ such that $\hat{\mathbb{Q}}_n \overset{d}{\to} \mu$. By the same argument before, we also have $\mathbb{E}_{X,X' \overset{iid}{\sim} \hat{\mathbb{Q}}_{k(n)}} U(|X - X'|) \to \mathbb{E}_{X,X' \overset{iid}{\sim} \mu} U(|X - X'|)$. Then by the optimal assumption of $\hat{\mathbf{r}}_n$ ,

$$\mathbb{E}_{X,X' \overset{iid}{\sim} \hat{\mu}} U(|X - X'|) = \lim_{n \to \infty} \mathbb{E}_{X,X' \overset{iid}{\sim} \hat{\mathbb{P}}_{k(n)}} U(|X - X'|)$$

$$\geq \lim_{n \to \infty} \mathbb{E}_{X,X' \overset{iid}{\sim} \hat{\mathbb{Q}}_{k(n)}} U(|X - X'|) = \mathbb{E}_{X,X' \overset{iid}{\sim} \mu} U(|X - X'|).$$

This means $\hat{\mu}$ maximize $\mathbb{E}_{X,X' \overset{iid}{\sim} \mu} U(|X - X'|)$ over all probability measure $\mu$ on $[0,1]$. From Lemma 4.1, we know that $1 - \hat{r}_i = \hat{r}_{n-i+1}$, so $\hat{\mu}$ is symmetric. If there is another sub-sequence $m(n)$ such that $\hat{\mathbb{P}}_{m(n)} \overset{d}{\to} \hat{\nu}$. By the same argument before, $\hat{\nu}$ is also optimal and symmetric. From Lemma 4.4, $\hat{\mu} = \hat{\nu}$. Thus for every converging sub-sequence of $\{\hat{\mathbb{P}}_n\}$, the limit distribution must be the same. By the tightness of $\{\hat{\mathbb{P}}_n\}$, we have $\hat{\mathbb{P}}_n \overset{d}{\to} \mu^*$. $\qquad\square$

### 4.2 Proof of Theorem 2

For the utility function $U(z) = z^\gamma$, having established Theorem 5, our objective is to identify a symmetric probability measure $\mu^*$ that maximizes $\mathbb{E}_{X,X' \overset{iid}{\sim} \mu} U(|X - X'|)$. By employing the variational principle, we can derive a condition that is necessary for optimality. Notably, this condition also suffices for optimality.

**Lemma 4.5.** *Let $U(z) = z^\gamma$ for some $\gamma \in (0,1)$. A probability measure $\mu$ on $[0,1]$ will maximize $\mathbb{E}_{X,X' \overset{iid}{\sim} \mu} U(|X - X'|)$ if it satisfies the condition that $\mathbb{E}_{X \sim \mu} U(|X - c|)$ is independent of $c \in [0,1]$.*

The proof of Lemma 4.5 is provided in Appendix D.1. Therefore, proving Theorem 2 is reduced to verifying the condition stated in Lemma 4.5. This verification process is tedious and will be deferred to Appendix D.2 for brevity.

## 5 Experiments

In this section, we conduct experiments to investigate the phenomenon of reward collapse and demonstrate that prompt-aware training can prevent reward collapse.

### 5.1 Evidence of reward collapse in Large language model

We start our investigation by conducting experiments utilizing a LLM, specifically GPT-Neo-1.3B (Black et al. (2021)). Guided by the methodologies outlined in the StackLlama project (Beeching et al. (2023)), we trained the model on the StackExchange preference dataset (Lambert et al. (2023)), a robust resource that provides rankings of responses for individual prompts.

Constrained by computational resources, we focused our training on a carefully selected subset of the dataset containing only the prompts accompanied by exactly five responses. Our experimental setup comprised 128 distinct prompts, each of which contributed 10 pairs to the reward modeling process. By adopting the codebase from StackLlama (Beeching et al. (2023)), and setting the learning rate to $3 \times 10^{-5}$ along with a batch size of 20 pairs, we carried out the training over 10 epochs.

As demonstrated in Figure 1, our results highlight the emergence of the reward collapse phenomenon under these realistic conditions. The evidence of this effect can be observed as the distribution becomes increasingly concentrated over the course of the training.

### 5.2 Setup of our second experiment

The open-source datasets currently available for RLHF are rather limited. Most of these datasets (Nakano et al. (2021); Bai et al. (2022a)) typically include only a handful of candidate responses (usually a single pair) for each corresponding prompt question. Moreover, the ranking signals in those datasets are usually noisy, either because they are sourced from the Internet (Ethayarajh et al. (2023)) or because of the inherent subjectivity of the ranking process.

In order to conduct a carefully controlled experiment, we curated our own dataset, focusing on a single, simplified feature – the length of the response, measured in terms of word count as the ground truth reward. A subset of questions was selected from the LongForm dataset (Köksal et al. (2023)), a question-answer dataset characterized by its lengthy answers. To simulate scenarios with open-ended and concrete problems, we truncated the original answer according to two distinct length distributions, thereby generating eight responses for each prompt: the first distribution is nearly uniform, ranging from 10 to 80 words, while the

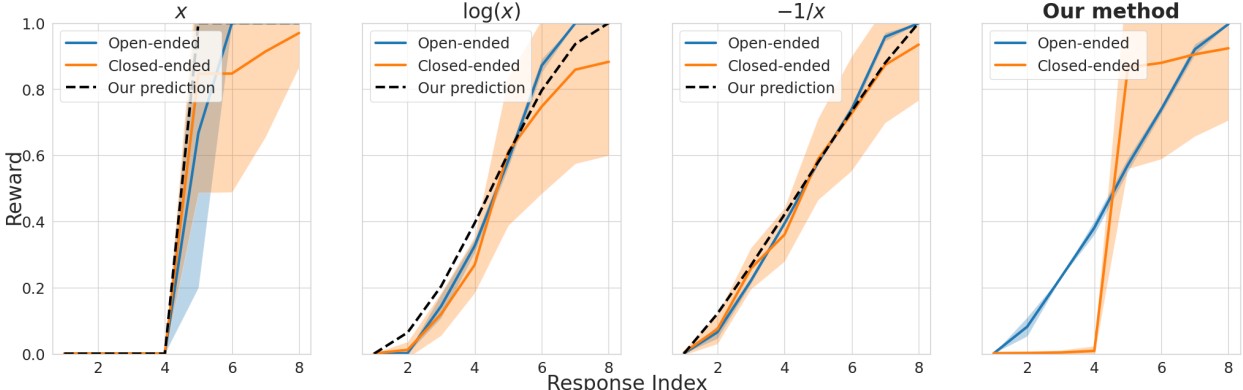

Figure 3: **Reward collapse on the test set.** The $x$-axis represents the response index, sorted by reward from smallest to largest, consistent with the following figure. The reward distributions exhibit similar collapse phenomena on the test set, and employing a prompt-aware loss function can mitigate this collapse.

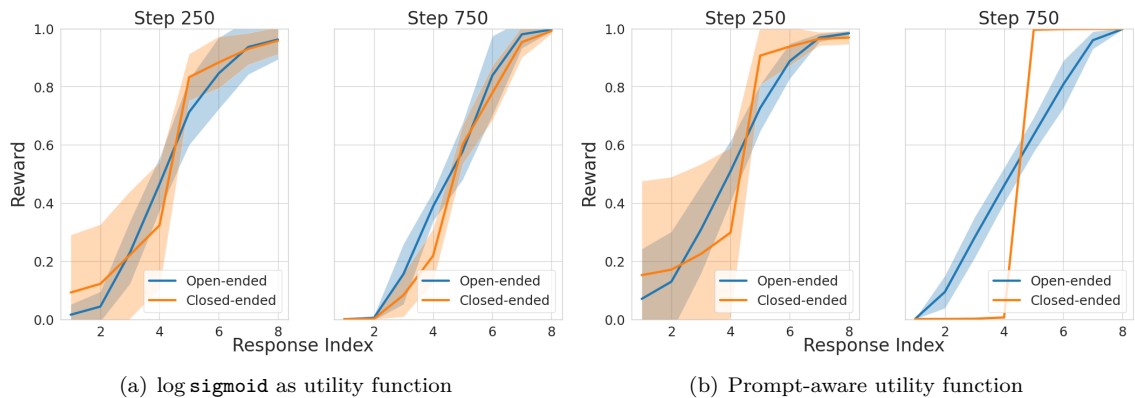

(a) log `sigmoid` as utility function      (b) Prompt-aware utility function

Figure 4: **(Left)** The reward distribution of different prompts gradually converges into a single distribution during training. **(Right)** When using the prompt-aware loss function, the reward distributions of the two different prompts can be gradually separated during training.

second is a polarized distribution with response lengths primarily clustered around either 30 or 60 words. Each question was randomly assigned as either open-ended or concrete. [4] Additionally, the phrases "Write the answer in an open-ended way." and "Write either a short answer or a long answer." were added to the open-ended and concrete questions, respectively, to distinguish the question type. Following this process, we constructed a dataset comprising 8192 training questions and 16 test questions.

In our experiments, we focus on the following $U$ functions: $z$, $\log z$, $-1/z$, as well as $\log \mathtt{sigmoid}(z)$, which is employed in Ouyang et al. (2022) and the prompt-aware $U$, which adaptively selects $U$ from $z$ and $-1/z$. Given that the $U$ function operates on $z$ in the range $[-1, 1]$, we adjust some $U$ functions with suitable continuous extensions or scaling. We then train a DeBERTa V3 (He et al. (2021)) as the reward model. The training details can be found in Appendix A.1.

## 5.3 Experimental results

**Fixed loss function leads to reward collapse.** As depicted in Figure 4(a), reward distributions corresponding to different prompts gradually converge towards a single, prompt-independent distribution throughout the training process. Specifically, in the context of Figure 4(a), where the $U$ function is represented by `LogSigmoid`, the reward distribution exhibits positive probability mass at reward scores of 0 and 1 (illus-

---

[4]In practice, such assignments can be done by various methods. See Appendix A.2 for a short discussion.

trated by the flat segments corresponding to the first two and last two scores). This observation validates the prediction encapsulated in Theorem 4. Examining other $U$ functions, Figures 3 collectively indicates the occurrence of loss collapse on the test datasets. Specifically, employing $z$ as the $U$ function results in a polarized reward distribution, whereas utilizing $-1/z$ as the $U$ function yields a uniform reward distribution.

**Prompt-aware training avoids reward collapse.** Figures 3 shows the reward distribution at the end of training with varying utility functions. The results along with Figure 4(b) reveal that using a prompt-aware $U$ function effectively prevents reward collapse across both training and test datasets. This strategy yields a more uniform reward distribution for open-ended prompts while promoting a more polarized reward distribution for concrete prompts.

## 6 Discussion

In this paper, we have introduced an empirical phenomenon known as reward collapse that arises during reward model training for aligning LLMs using human preference rankings. This phenomenon results in the same reward distribution regardless of the prompt type. The occurrence of reward collapse stems from neural network interpolation during the final training phase. Although techniques that mitigate overfitting, such as early stopping or regularization, can be employed to mitigate reward collapse, we propose a new method that consider the nature of prompts. We provided an analytical framework that evaluates reward distribution, yielding closed-form reward expressions. Synthetic experiments substantiate our findings, presenting a method superior to early stopping to tackle reward collapse.

While our experiments provide valuable insights, it is important to acknowledge their limitations, primarily stemming from the constrained computational resources available. Given abundant resources, future research can explore the use of a more diverse range of prompts, varying in terms of their open-endedness. Additionally, it would be interesting to investigate the extent to which the trained reward model enhances the capabilities of large language models, such as their ability to self-calibrate uncertainty (Lin et al. (2022); Kadavath et al. (2022)). Theoretical investigations could focus on finding increasing, concave functions that precisely match a given discrete reward distribution. On the practical side, developing a method to choose a utility function based on prompts, perhaps using a parameter such as $\gamma$ in Section 3, poses an intriguing avenue for further exploration (more discussion on choosing utility function is in Appendix A.2). Furthermore, exploring the potential benefits of truncated ranking by requiring human labelers to provide partial rankings of acceptable completions and ignore unacceptable completions could offer valuable insights into improving the training of reward models.

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

# A  Details about experiments

## A.1  Training Details

We use the following extension of the utility functions during our training.

- $\log z$: $U(z) = \begin{cases} \log(z + \epsilon) & \text{for } z > 0 \\ z + \log(\epsilon) & \text{for } z \leq 0 \end{cases}$, where $\epsilon$ is set to 0.1.

- $-1/z$: $U(z) = \begin{cases} -1/(z + \epsilon) & \text{for } z > 0 \\ z - 1/\epsilon & \text{for } z \leq 0 \end{cases}$, where $\epsilon$ is also set to 0.1.

- $\log \texttt{sigmoid}(z)$: $U(z) = \log \texttt{sigmoid}(4z)$. Here, the scaling factor of 4 ensures the output of $\log \texttt{sigmoid}$ spans a sufficient range.

To train the reward model, we adopted the approach used in the OpenAssistant project, which utilizes the DeBERTaV3 Base model He et al. (2021). To constrain the reward output between 0 and 1, a $\sigma$ function was appended before the final output. The reward model was trained with a batch size of 224 (comprising eight questions per batch, each with 28 pairs) for a total of 1000 steps, approximately equivalent to 1 epoch. The maximum learning rate was configured to 1e-5, utilizing the Adam optimizer and a linear learning rate schedule, inclusive of 10% warmup steps. The reward model was trained on a single A6000 GPU, with the entire training process concluding in roughly 1 hour.

## A.2  Discussion on assigning the prompt type

Determining the prompt type is a crucial aspect of our prompt-aware approach. In our experiments, we randomly assigned prompts as either open-ended or close-ended. This sufficed to demonstrate the effectiveness of our prompt-aware approach in shaping the reward distribution for different types of prompts. However, in practice, there are various viable methods to accomplish this. While our primary focus is not on detailing how to identify the prompt type, we intend to present some straightforward yet effective approaches.

One straightforward method is manually deciding the prompt type, similar to how human feedback is collected in Instructgpt (Ouyang et al. (2022)), where preference rankings are collected from human labelers. Typically, these labelers are asked to rank different responses. Moreover, we can ask them to evaluate the extent of open-endedness in the prompt, using a scale that ranges from -1 to 1.

Automated annotation processes are also possible. For example, one could assess the diversity of responses to a given prompt. If the responses exhibit significant diversity, the prompt could be categorized as open-ended. Conversely, if the responses show limited diversity, the prompt might be classified as close-ended.

Determining the prompt type is indeed a complex and intriguing task, and it offers an interesting avenue for future research.

# B  Proofs of Theorem 1

*Proof.* When $U$ is $\mu$-strong concave, by Lemma 4.2 and Lemma 4.4, $S$ defined in (1) has a unique maximizer $\hat{r}_1, \cdots, \hat{r}_n$. Moreover, for any $u_1, \cdots, u_n$,

$$S(u_1, \cdots, u_n) \leq S(\hat{r}_1, \cdots, \hat{r}_n) - \frac{n\mu}{2} \| \text{Proj}_{V_n}(\mathbf{u} - \hat{\mathbf{r}}) \|^2.$$

Here, $V_n \subset \mathbb{R}^n$ is the subspace orthogonal to $(1, \cdots, 1)$, and $\| \cdot \|$ represents the Euclidean norm. Back to the Theorem 1, if a neural network $R_\theta$ satisfies

$$\sum_{(x, y_1, \cdots, y_n) \in \mathcal{D}} S(R_\theta(x, y_1), \cdots, R_\theta(x, y_n)) \geq |\mathcal{D}|M - \frac{\mu n \epsilon^2}{2},$$

then for all $(x, y_1, \cdots, y_n) \in \mathcal{D}$, $S(R_\theta(x, y_1), \cdots, R_\theta(x, y_n)) \geq M - \frac{\mu n \epsilon^2}{2}$ because the maximum of $S$ is $M$. As a result, letting $\mu = ((R_\theta(x, y_1), \cdots, R_\theta(x, y_n)))$,

$$M - \frac{\mu n \epsilon^2}{2} \leq S(R_\theta(x, y_1), \cdots, R_\theta(x, y_n)) \leq M - \frac{n\mu}{2} \| \operatorname{Proj}_{V_n}(\mathbf{u} - \hat{\mathbf{r}}) \|^2.$$

This gives an upper bound $\epsilon^2$ on $\| \operatorname{Proj}_{V_n}(\mathbf{u} - \hat{\mathbf{r}}) \|^2$. Finally, by the definition of $\operatorname{Proj}_{V_n}$, there exists a constant $c(x)$, such that $\operatorname{Proj}_{V_n}(\mathbf{u} - \hat{\mathbf{r}}) = u - \hat{r} + c(x) \cdot \mathbf{1}$. For this constant $c(x)$,

$$\max_i |R_\theta(x, y_i) - \hat{r}_i - c(x)| \leq \sqrt{\| \operatorname{Proj}_{V_n}(\mathbf{u} - \hat{\mathbf{r}}) \|^2} \leq \epsilon.$$

This finishes the proof. $\qquad\qquad\square$

## C  Missing Proofs in Section 2 and 4

### C.1  Proof of Lemma 4.1

We break the proof of Lemma 4.1 into two different lemma.

**Lemma C.1.** *If the utility function $U(x)$ is strictly increasing, let $\hat{\mathbf{r}}$ be the solution of optimization problem:*

$$\max_{0 \leq r_1, \ldots, r_n \leq 1} \sum_{1 \leq i < j \leq n} U(r_i - r_j)$$

*Then $\hat{\mathbf{r}}$ satisfies: $\hat{r}_1 \geq \cdots \geq \hat{r}_n$.*

*Proof.* Let $S(\mathbf{r}) = \sum_{1 \leq i < j \leq n} U(r_i - r_j)$. Suppose the conclusion is not true, then there exists a $k \geq 0$, such that $\hat{r}_1 \geq \cdots \geq \hat{r}_k$ and $\hat{r}_k < \hat{r}_{k+1}$. Let us define

$$\tilde{r}_i = \begin{cases} \hat{r}_i & \text{if } i \neq k, k+1; \\ \hat{r}_{k+1} & \text{if } i = k; \\ \hat{r}_k & \text{if } i = k+1. \end{cases}$$

Then

$$\sum_{1 \leq i < j \leq n} U(\hat{r}_i - \hat{r}_j) - \sum_{1 \leq i < j \leq n} U(\tilde{r}_i - \tilde{r}_j) = U(\hat{r}_k - \hat{r}_{k+1}) - U(\hat{r}_{k+1} - \hat{r}_k) < 0$$

because $U$ is strictly increasing and $\hat{r}_k - \hat{r}_{k+1} < 0$. This contracts with the fact that $\hat{\mathbf{r}}$ is the solution of the optimization problem, and thus the conclusion holds. $\qquad\square$

**Lemma C.2.** *If the utility function $U(x)$ is strictly increasing and strictly concave, then the function $S(\mathbf{r}) = \sum_{1 \leq i < j \leq n} U(r_i - r_j)$ is concave. Moreover, the solution of optimization problem*

$$\max_{0 \leq r_1, \ldots, r_n \leq 1} \sum_{1 \leq i < j \leq n} U(r_i - r_j)$$

*is unique and satisfies: $1 - \hat{r}_i = \hat{r}_{n-i+1}$ for $i = 1, 2, \cdots, n$.*

*Proof.* The concavity of $S$ follows directly from definition:

$$S(\mathbf{r}) + S(\mathbf{r}') = \sum_{1 \leq i < j \leq n} U(r_i - r_j) + U(r_i' - r_j')$$

$$\leq \sum_{1 \leq i < j \leq n} 2U(\frac{r_i + r_i' - r_j - r_j'}{2}) = 2S(\frac{\mathbf{r} + \mathbf{r}'}{2}).$$

The above inequality is an equality if and only if $r_i - r_j = r'_i - r'_j$ for all $1 \leq i < j \leq n$ when $U(x)$ is strictly concave. When $U$ is increasing, the solution $\hat{\mathbf{r}}$ of the optimization problem satisfies $\hat{r}_1 = 1$. Thus the solution of the optimization problem $\max_{1 \leq r_1, \cdots, r_n \leq 1} S(\mathbf{r})$ is unique, otherwise the vector $\frac{\mathbf{r}_1 + \mathbf{r}_2}{2}$ makes $S$ larger where $\mathbf{r}_1$ and $\mathbf{r}_2$ are two different solutions.

Finally, let $\hat{\mathbf{r}}$ be the unique solution of the optimization problem. Let us define $\tilde{r}_i = 1 - \hat{r}_{n-i+1}$ for all $i = 1, 2, \cdots, n$. It follows that $\tilde{r}_i - \tilde{r}_j = \hat{r}_{n-j+1} - \hat{r}_{n-i+1}$, and we have $S(\hat{\mathbf{r}}) = S(\tilde{\mathbf{r}})$. Consequently, the uniqueness of the solution implies $\hat{\mathbf{r}} = \tilde{\mathbf{r}}$. This means that $\hat{r}_i = 1 - \hat{r}_{n-i+1}$ for $i = 1, \cdots, n$. □

## C.2 Proof of Lemma 4.2

*Proof of Lemma 4.2.* The definition of $S(\mathbf{r})$ is

$$S(\mathbf{r}) = \sum_{1 \leq i < j \leq n} U(r_i - r_j).$$

The value of $S$ does not change if we increase all $r_i$ by the same constant. Thus the value of $S(\mathbf{r})$ only depends on $\mathrm{Proj}_{V_n}(\mathbf{r})$ where $V_n \subset \mathbb{R}^n$ denotes the subspace orthogonal to $(1, 1, \cdots, 1)$. We can define a new function on $V_n$ by letting

$$F(\mathrm{Proj}_{V_n}(\mathbf{r})) = S(\mathbf{r}).$$

The domain of $F$ is $A = \{\mathbf{v} \in V_n | \exists \mathbf{r} \in \mathbb{R}^n \text{ such that } 0 \leq r_i \leq 1 \text{ and } \mathbf{v} = \mathrm{Proj}_{V_n}(\mathbf{r})\}$. First, we can show that $F$ is $n\mu$-strongly concave.

Because $U$ is $\mu$-strongly concave, $U(x) + \frac{\mu}{2}x^2$ is concave. It follows that

$$S(\mathbf{r}) + \frac{\mu}{2} \sum_{1 \leq i < j \leq n} (r_i - r_j)^2$$

is also concave. We can write $\sum_{1 \leq i < j \leq n} (r_i - r_j)^2$ as

$$\sum_{1 \leq i < j \leq n} (r_i - r_j)^2 = n \sum_{i=1}^{n} r_i^2 - \left(\sum_{i=1}^{n} r_i\right)^2$$

by Lagrange identity. Then note that $V_n$ is the subspace orthogonal to $(1, 1, \cdots, 1)$. The projection onto $V_n$ is given by

$$\mathrm{Proj}_{V_n}(\mathbf{r}) = \left(r_1 - \frac{1}{n} \sum_{i=1}^{n} r_i, \cdots, r_n - \frac{1}{n} \sum_{i=1}^{n} r_i\right).$$

As a result,

$$\|\mathrm{Proj}_{V_n}(\mathbf{r})\|^2 = \sum_{i=1}^{n} \left(r_i - \frac{1}{n} \sum_{j=1}^{n} r_j\right)^2 = \sum_{i=1}^{n} r_i^2 - \frac{1}{n}\left(\sum_{i=1}^{n} r_i\right)^2 = \frac{1}{n} \sum_{1 \leq i < j \leq n} (r_i - r_j)^2.$$

From this equation and the concavity of $S(\mathbf{r}) + \frac{\mu}{2} \sum_{1 \leq i < j \leq n} (r_i - r_j)^2$, we know that

$$S(\mathbf{r}) + \frac{n\mu}{2} \|\mathrm{Proj}_{V_n}(\mathbf{r})\|^2$$

is also concave. Consequently, $F(\mathrm{Proj}_{V_n}(\mathbf{r})) + \frac{n\mu}{2} \|\mathrm{Proj}_{V_n}(\mathbf{r})\|^2$ is concave, which lead to the strong concavity of $F$ because

$$F(\mathbf{v}) + \frac{n\mu}{2} \|\mathbf{v}\|^2$$

is concave. Let $\hat{\mathbf{v}}$ be the optimal vector that maximizes $F(\mathbf{v})$, strong concavity implies (See e.g. Section 9.1.2 in Boyd & Vandenberghe (2004))

$$F(\mathbf{v}) - F(\hat{\mathbf{v}}) \leq -\frac{n\mu}{2}\|\mathbf{v} - \hat{\mathbf{v}}\|^2.$$

Therefore, by the definition of $F(\text{Proj}_{V_n}(\mathbf{r})) = S(\mathbf{r})$, we have

$$S(\mathbf{u}) - S(\hat{\mathbf{r}}) \leq -\frac{n\mu}{2}\|\text{Proj}_{V_n}(\mathbf{u} - \hat{\mathbf{r}})\|^2.$$

$\square$

### C.3 Proof of Lemma 4.3

*Proof of Lemma 4.3.* Because $\mu_j, j = 1, 2$ are symmetric, we have

$$
\begin{aligned}
r^{(j)}_{n,n-i+1} &= \frac{1}{2}\inf\{t : \mu_j([0,t]) \geq \frac{i-1}{n-1}\} + \frac{1}{2}\sup\{t : \mu_j([0,t)) < \frac{i-1}{n-1}\}\}) \\
&= \frac{1}{2}(1 - \sup\{t : \mu_j([t,1]) \geq \frac{i-1}{n-1}\}) + \frac{1}{2}(1 - \inf\{t : \mu_j((t,1]) < \frac{i-1}{n-1}\}) \\
&= \frac{1}{2}(1 - \sup\{t : \mu_j([0,t)) < \frac{n-i}{n-1}\}) + \frac{1}{2}(1 - \inf\{t : \mu_j([0,t]) \geq \frac{n-i}{n-1}\} \\
&= 1 - r^{(j)}_{n,i}.
\end{aligned}
$$

So we have $\sum_{i=1}^{n}(r^{(1)}_{n,i} - r^{(2)}_{n,i}) = 0$. Note that $V_n \subset \mathbb{R}^n$ is the subspace which is orthogonal to $(1, 1, \cdots, 1)$, the projection of $\mathbf{x} = (x_1, \cdots, x_n)$ onto $V_n$ is given by

$$\text{Proj}_{V_n}(\mathbf{x}) = (x_1 - \frac{1}{n}\sum_{i=1}^{n}x_i, \cdots, x_n - \frac{1}{n}\sum_{i=1}^{n}x_i).$$

Consequently,

$$\|\text{Proj}_{V_n}(\mathbf{r}^{(1)}_n - \mathbf{r}^{(2)}_n)\|_2^2 = \sum_{i=1}^{n}(r^{(1)}_{n,i} - r^{(2)}_{n,i})^2.$$

If $\mu_1$ and $\mu_2$ are two different symmetric probability measure on $[0,1]$, we can assume that there exists $q_1 < q_2 \in [0,1]$ and $\delta \geq 0$, such that $\mu_1([0,q_2]) < \mu_2([0,q_1]) - \delta$. So when $\frac{i-1}{n-1} \in (\mu_1([0,q_2]), \mu_2([0,q_1]) - \delta)$, we have $r^{(1)}_{n,n-i+1} \geq q_2$ because $\mu_1([0,q_2]) < \frac{i-1}{n-1}$. We also have $r^{(2)}_{n,n-i+1} \leq q_1$ because $\mu_2([0,q_1]) > \frac{i-1}{n-1}$. As a result, $r^{(1)}_{n,n-i+1} - r^{(2)}_{n,n-i+1} \geq q_2 - q_1$ whenever $(i-1)/(n-1) \in (\mu_1([0,q_2]), \mu_2([0,q_1]) - \delta)$. Because the length of the interval is positive, the number of such $i$ is larger than $c_1 n$ where $c_1$ is a constant independent of $n$. Then we conclude that

$$
\begin{aligned}
\|\text{Proj}_{V_n}(\mathbf{r}^{(1)}_n - \mathbf{r}^{(2)}_n)\|_2^2 &= \sum_{i=1}^{n}(r^{(1)}_{n,i} - r^{(2)}_{n,i})^2 \\
&\geq c_1 n(q_1 - q_2)^2.
\end{aligned}
$$

Choosing $c_0 = c_1(q_1 - q_2)^2$ gives the inequality

$$\|\text{Proj}_{V_n}(\mathbf{r}^{(1)}_n - \mathbf{r}^{(2)}_n)\|_2^2 \geq c_0 n.$$

$\square$

### C.4 Proof of Lemma 4.4

*Proof of Lemma 4.4.* Suppose there exist two different symmetric probability measure $\mu_1$ and $\mu_2$, they both maximize $\mathbb{E}_{X,X'\overset{iid}{\sim}\mu} U(|X - X'|)$. Let $M = \mathbb{E}_{X,X'\overset{iid}{\sim}\mu_j} U(|X - X'|), j = 1, 2$. Now let $r_{n,i}^{(j)} = \frac{1}{2}\inf\{t :$ $\mu_j([0,t]) \geq \frac{i-1}{n-1}\} + \frac{1}{2}\sup\{t : \mu_j([0,t)) < \frac{i-1}{n-1}\}\}), i = 1, 2, \cdots, n; j = 1, 2$ as defined in Lemma 4.3. Accordingly, let $\mathbb{P}_n^{(j)} = \frac{1}{n}\sum_{i=1}^n \delta_{r_{n,i}^{(j)}}$. Then we have

$$\mathbb{P}_n^{(j)} \overset{d}{\to} \mu_j, j = 1, 2.$$

This can be proved easily by considering the definition of convergence in distribution. Since $G$ is bounded, this lead to $\mathbb{E}_{X,X'\overset{iid}{\sim}\mathbb{P}_n^{(j)}} U(|X - X'|) \to M, j = 1, 2$ as $n \to \infty$.

The expectation $\mathbb{E}_{X,X'\overset{iid}{\sim}\mathbb{P}_n^{(j)}} U(|X - X'|)$ can be written more precisely as

$$\mathbb{E}_{X,X'\overset{iid}{\sim}\mathbb{P}_n^{(j)}} U(|X - X'|) = \frac{1}{n^2}\sum_{1 \leq i,i' \leq n} U(|r_{n,i}^{(j)} - r_{n,i'}^{(j)}|).$$

By Lemma 4.2, we can bound the difference

$$\frac{1}{n^2}\sum_{1 \leq i,i' \leq n} U(|r_{n,i}^{(j)} - r_{n,i'}^{(j)}|) - \frac{1}{n^2}\sum_{1 \leq i \leq i' \leq n} U(|\hat{r}_{n,i} - \hat{r}_{n,i'}|)$$

$$= \frac{2}{\binom{n}{2}}\sum_{1 \leq i < i' \leq n} U(r_{n,i}^{(j)} - r_{n,i'}^{(j)}) - \frac{2}{\binom{n}{2}}\sum_{1 \leq i < i' \leq n} U(\hat{r}_{n,i} - \hat{r}_{n,i'})$$

$$\leq -\frac{2\mu}{n-1}\|\operatorname{Proj}_{V_n}(\mathbf{r}_n^{(j)} - \hat{\mathbf{r}}_n)\|_2^2.$$

Then apply Lemma 4.3, there exist $c_0 \geq 0$ such that

$$2\|\operatorname{Proj}_{V_n}(\mathbf{r}_n^{(1)} - \hat{\mathbf{r}}_n)\|^2 + 2\|\operatorname{Proj}_{V_n}(\mathbf{r}_n^{(2)} - \hat{\mathbf{r}}_n)\|^2 \geq \|\operatorname{Proj}_{V_n}(\mathbf{r}_n^{(1)} - \mathbf{r}_n^{(2)})\|^2.$$

Here, we uses $2\|x\|_2^2 + 2\|y\|_2^2 \geq \|x - y\|_2^2$. So

$$\min_{j=1,2} \frac{1}{n^2}\left[\sum_{1 \leq i,i' \leq n} U(|r_{n,i}^{(j)} - r_{n,i'}^{(j)}|) - U(|\hat{r}_{n,i} - \hat{r}_{n,i'}|)\right]$$

$$= -\frac{2\mu}{n-1}\max_{j=1,2}\|\operatorname{Proj}_{V_n}(\mathbf{r}_n^{(j)} - \hat{\mathbf{r}}_n)\|_2^2$$

$$\leq -\frac{2\mu}{n-1}\frac{\|\operatorname{Proj}_{V_n}(\mathbf{r}_n^{(1)} - \mathbf{r}_n^{(2)})\|^2}{4}$$

$$\leq -\frac{\mu c_0 n}{2n-2} \leq -\frac{c_0\mu}{2}.$$

Since $M = \max \mathbb{E}_{X,X'\overset{iid}{\sim}\mu} U(|X - X'|)$, we know $\frac{1}{n^2}\sum_{1 \leq i,i' \leq n} U(|\hat{r}_{n,i} - \hat{r}_{n,i'}|) \leq M$. As a result,

$$\min_{j=1,2} \mathbb{E}_{X,X'\overset{iid}{\sim}\mathbb{P}_n^{(j)}} U(|X - X'|) \leq \frac{1}{n^2}\sum_{1 \leq i \leq i' \leq n} U(|\hat{r}_{n,i} - \hat{r}_{n,i'}|) - \mu c_0/2 \leq M - \mu c_0/2.$$

This contradicts the assumption that $\mathbb{E}_{X,X'\overset{iid}{\sim}\mathbb{P}_n^{(j)}} U(|X - X'|) \to M, j = 1, 2, \ n \to \infty$.

$\square$

# D Proof of Theorem 2

Given Theorem 5, we only need to find a symmetric probability measure on $[0, 1]$, which maximizes

$$\mathbb{E}_{X, X' \overset{iid}{\sim} \mu} U(|X - X'|).$$

The following proof in this section is adapted from (https://math.stackexchange.com/users/9340/sangchul lee). Let $M(\mathcal{B}([0, 1]))$ denote the sets of all finite signed measure on the Borel sigma algebra $\mathcal{B}([0, 1])$. Apparently, $P(\mathcal{B}([0, 1])) \subset M(\mathcal{B}([0, 1]))$. Then we define the following "inner product" in $M(\mathcal{B}([0, 1]))$:

$$\langle \mu, \nu \rangle = \mathbb{E}_{X \sim \mu, X' \sim \nu, \text{independent}} U(|X - X'|) = \int_{[0,1]^2} U(|x - y|) \mu(dx) \nu(dy).$$

We also define $I(\mu)$ as $I(\mu) := \langle \mu, \mu \rangle$. With these notations, the problem becomes

$$\max_{\mu \in P(\mathcal{B}([0,1]))} I(\mu).$$

**Lemma D.1.** *For $U(x) = x^\gamma$ with $\gamma \in (0, 1)$. If $\mu$ is a signed measure satisfying $\mu([0, 1]) = 0$, then we have $I(\mu) \leq 0$. Moreover, $I(\mu) = 0$ if and only if $\mu(E) = 0$ for all $E \subset [0, 1]$.*

*Proof.* $f(t) = \frac{1 - \cos(xt)}{t^{1+\gamma}}$ is integrable on $(0, \infty)$. As a result, using change of variables, we have

$$|x|^\gamma = C \int_0^\infty \frac{1 - \cos(xt)}{t^{1+\gamma}} dt$$

for come constant $C > 0$. Then by Fubini's theorem, we have

$$\begin{aligned}
\langle \mu, \mu \rangle &= \int_{[0,1]^2} |x - y|^\gamma \mu(dx) \mu(dy) \\
&= C \int_{[0,1]^2} \int_0^\infty \frac{1 - \cos((x-y)t)}{t^{1+\gamma}} dt \mu(dx) \mu(dy) \\
&= C \int_0^\infty \left( \int_{[0,1]^2} \frac{1 - \cos((x-y)t)}{t^{1+\gamma}} \mu(dx) \mu(dy) \right) dt.
\end{aligned}$$

Note that $\cos((x - y)t) = \Re(e^{ixt - iyt})$, we have

$$\begin{aligned}
\int_{[0,1]^2} & \frac{1 - \cos((x-y)t)}{t^{1+\gamma}} \mu(dx) \mu(dy) \\
&= -\Re \left( \int_{[0,1]^2} \frac{e^{ixt} e^{-iyt}}{t^{1+\gamma}} \mu(dx) \mu(dy) \right) \\
&= -\Re \left( |\hat{\mu}(t)|^2 \right),
\end{aligned}$$

where $\hat{\mu}(t) = \int_{[0,1]} e^{itx} \mu(dx)$ is the Fourier transform of $\mu$. Then

$$I(\mu) = -C \int_0^\infty \frac{|\hat{\mu}(t)|^2}{t^{1+\gamma}} dt \leq 0.$$

Moreover, $I(\mu) = 0$ if and only if $\hat{\mu}(t) = 0$ for all $t \in [0, \infty)$ if and only if $\mu(E) = 0$ for all $E \in \mathcal{B}([0, 1])$. $\square$

## D.1 Proof of Lemma 4.5

We first restate the lemma.

**Lemma D.2.** *Let $U(x) = x^\gamma$ for some $\gamma \in (0, 1)$. If a probability measure $\mu$ on $[0, 1]$ maximize $\mathbb{E}_{X, X' \overset{iid}{\sim} \mu} U(|X - X'|)$ if it satisfies that $\mathbb{E}_{X \sim \mu} U(|X - c|)$ does not depend on $c \in [0, 1]$.*

*Proof of Lemma 4.5.* For two probability measure $\mu$ and $\nu$ on $[0,1]$, $(\mu - \nu)([0,1]) = 0$. Suppose $\mu$ satisfies $\mathbb{E}_{X \sim \mu} U(|X - c|) = K$ does not depend on $c \in [0,1]$. Note that

$$\langle \nu - \mu, \mu \rangle = \int_{[0,1]} \left( \int_{[0,1]} |x - y|^\gamma \mu(dx) \right) (\nu - \mu)(dy) = \int_{[0,1]} K(\nu - \mu)(dy) = 0.$$

And by lemma D.1, $\langle \nu - \mu, \nu - \mu \rangle \leq 0$. Therefore,

$$\langle \nu, \nu \rangle = \langle \mu, \mu \rangle + 2\langle \nu - \mu, \mu \rangle + \langle \nu - \mu, \nu - \mu \rangle \leq \langle \mu, \mu \rangle.$$

This means that $\mu$ maximize $\mathbb{E}_{X, X' \overset{iid}{\sim} \mu} U(|X - X'|)$. $\qquad\qquad\square$

## D.2 Proof of Theorem 2

*Proof of Theorem 2.* Let $\mu$ be the probability measure induced by $\text{Beta}(\frac{1-\gamma}{2}, \frac{1-\gamma}{2})$. It has probability density function

$$f_\gamma(x) = \frac{1}{B(\frac{1-\gamma}{2}, \frac{1-\gamma}{2})} x^{-\frac{1+\gamma}{2}} (1-x)^{-\frac{1+\gamma}{2}}.$$

For any $c \in [0,1]$, $\mathbb{E}_{X \sim \mu} U(|X - c|)$ can be expressed as

$$
\begin{aligned}
\mathbb{E}_{X \sim \mu} U(|X - c|) &= \frac{1}{B(\frac{1-\gamma}{2}, \frac{1-\gamma}{2})} \int_0^1 |x - c|^\gamma x^{-\frac{1+\gamma}{2}} (1-x)^{-\frac{1+\gamma}{2}} dx \\
&= \frac{1}{B(\frac{1-\gamma}{2}, \frac{1-\gamma}{2})} \int_0^{\frac{\pi}{2}} |\sin^2\theta - c|^\gamma (\sin\theta)^{-1-\gamma} (\cos\theta)^{-1-\gamma} d\sin^2\theta \\
&= \frac{2}{B(\frac{1-\gamma}{2}, \frac{1-\gamma}{2})} \int_0^{\frac{\pi}{2}} \left| \frac{\sin^2\theta - c}{\sin\theta\cos\theta} \right|^\gamma d\theta \\
&= \frac{2}{B(\frac{1-\gamma}{2}, \frac{1-\gamma}{2})} \int_0^\infty \left( \int_0^{\pi/2} \mathbf{1}\left\{ \left| \frac{\sin^2\theta - c}{\sin\theta\cos\theta} \right|^\gamma \geq t \right\} d\theta \right) dt.
\end{aligned}
$$

Because

$$
\begin{aligned}
\int_0^{\pi/2} \mathbf{1}\left\{ \left| \frac{\sin^2\theta - c}{\sin\theta\cos\theta} \right|^\gamma \geq t \right\} d\theta &= \frac{1}{2} \int_0^\pi \mathbf{1}\left\{ \left| \frac{\cos\theta + 2c - 1}{\sin\theta} \right|^\gamma \geq t \right\} d\theta \\
&= \frac{\pi}{2} - \frac{1}{2} \int_0^\pi \mathbf{1}\left\{ -\cos\theta - t^{1/\gamma}\sin\theta \leq 2c - 1 \leq -\cos\theta + t^{1/\gamma}\sin\theta \right\} d\theta \\
&= \frac{\pi}{2} - \frac{1}{2} \int_0^\pi \mathbf{1}\left\{ -\cos(\theta - \phi) \leq \frac{2c-1}{\sqrt{1 + t^{2/\gamma}}} \leq -\cos(\theta + \phi) \right\} d\theta \\
&= \frac{\pi}{2} - \phi,
\end{aligned}
$$

where $\tan\phi = t^{1/\gamma}$ and $\phi \in [0, \pi/2]$, and the last equation use the fact that $c \in [0,1]$. As a result, $\mathbb{E}_{X \sim \mu} U(|X - c|)$ does not depend on $c$.

Note that Beta distribution is also symmetric. It follows from Theorem 5 that the reward distribution converges to $\text{Beta}\left(\frac{1-\gamma}{2}, \frac{1-\gamma}{2}\right)$. $\qquad\qquad\square$

## D.3 Proof of Theorem 4

Theorem 4 can be intuitively understood as follows: If the function $U$ satisfies $U'(0) < \infty$ and $U'(1) > 0$, we can show, by analyzing the first-order optimality condition, that a positive fraction of $\hat{r}$ is equal to 1.

*Proof of Theorem 4.* The derivative of $-\sum_{i<j} U(r_i - r_j)$ with respect to $r_k$ is given by

$$-\left. \frac{\partial \sum_{i<j} U(r_i - r_j)}{\partial r_k} \right|_{\hat{r}_1, \cdots, \hat{r}_n} = \sum_{i=1}^{k-1} U'(\hat{r}_i - \hat{r}_k) - \sum_{i=k+1}^N U'(\hat{r}_k - \hat{r}_j) \leq (k-1)U'(0) - (n-k)U'(1).$$

The inequality follows from the convexity of $U$. Let $\kappa = \frac{U'(0)}{U'(1)}$. If $k \leq n/(\kappa+1)$, we have $(k-1)U'(0) - (n-k)U'(1) \leq 0$. Hence, we can get $\hat{r}_k = 1$. Otherwise, we could increase $\hat{r}_k$ to make $\sum_{i<j} U(\hat{r}_i - \hat{r}_j)$ larger. As a result, $\hat{r}_1 = \cdots = \hat{r}_{[n/(\kappa+1)]} = 1$. This gives $\hat{\mathbb{P}}_n(\{1\}) \geq [\frac{n}{\kappa+1}]/n$. By Theorem 5, we know that there exists a limiting distribution $\mu^*$ such that $\hat{\mathbb{P}} \xrightarrow{d} \mu^*$ and $\mu^*(\{1\}) \geq 1/(\kappa+1)$. Due to symmetry proved in Lemma 4.1, we also have $\mu^*(\{0\}) \geq 1/(\kappa+1)$. $\qquad\square$

