# OpenReview forum: "Reward Collapse in Aligning Large Language Models"
_TMLR — Rejected by TMLR_

### Review · Reviewer_Bse6 · 2024-08-05

**Summary Of Contributions:**

This paper identifies the reward collapse problem when training the reward model for aligning LLMs. The authors argue that this is caused by the insufficiency of the ranking-based objective to incorporate prompt-related information during optimization. The authors propose closed-form expressions for the reward utility function and show that such prompt-aware optimization scheme avoids the reward collapse problems in training reward models.

**Audience:**

Yes

**Broader Impact Concerns:**

No concern.

**Claims And Evidence:**

Yes

**Requested Changes:**

1. Could the authors provide the details in Section 3 on how the newly proposed utility functions are prompt aware? How do we get from the input sequence from the $\gamma$ variable?


2. For all the figures in the paper, please provide detailed descriptions in the paper. It is hard to get what the authors are trying to convery from the figure without any explanation on the results.


3. Provide results that how the new reward models affect the alignment of LLMs. I am curious as to whether it leads to better alignment.

**Strengths And Weaknesses:**

Strengths:
This paper tackles an important problem as how to better align large language models. Given that the alignment procedure is widely used in production LLMs, the topic this paper investigates should be of practical interest to the broad research community.


Weaknesses:
1. This might be a minor issue. But it is not clear to me what does “x” mean throughout the paper. In equation (4), “x” refers to the input sequence. However, in later parts of the paper which uses U(x), “x” means the input to the utility function. It would be good to clarify this inconsistencies in the paper.


2. In section 3, how exactly does the input sequence “x” affect the utility function? The input sequence “x”  and the hyper-parameter $\gamma$ in function “U” do not look as the same since the later is obviously a number. How are these two variables related? This seems an important aspect of the proposed approach and I couldn’t find any explanation in section 3.


3. Many important details regarding the experimental results are missing. For example, what exactly does Figure 1 represent? What does the x-axis “Response Index” mean? The authors mentioned in the caption of Figure 1 that “experiment details are elaborated in Section 5”. However, it it still missing in Section 5 the descriptions on how Figure 1 demonstrate the reward collapse phenomenon.


4. One important point that is missing is that the authors did not show whether the learnt reward model in this paper would actually lead to better alignment. It is good to see that it solves the reward collapse problem. But how are the alignment process affected by the new reward model? The paper should provide experimental analysis as this is a paper on aligning LLMs.

---

> ### Author Response · Authors · 2024-08-17
> **Response to Reviewer Bse6**
>
> Thank you for your valuable review of our paper. We would like to address some of your concerns as follows:
>
> 1. Clarification on the notation "x"
>
> We acknowledge that the overuse of the notation "x" throughout our paper may have caused confusion. To address this, we have revised the notation. Specifically, $x$ will now be used exclusively to denote the input sequence, while $z$ will represent the input to the utility function. Additionally, in our prompt-aware optimization, we use $U_x$ to denote a specific utility function that depends on the input sequence $x$. To avoid further confusion, we have removed the notation $U_\gamma$.
>
> 2. Impact of the input sequence "x" on the utility function (Section 3)
>
> This concern relates closely to the previous one. Our proposed prompt-aware optimization framework is intentionally general, allowing for any utility function $U_x$ that depends on $x$. In this paper, we investigate a specific utility function, $U_x(z) = z^{\gamma(x)}$, where $\gamma$ is a function mapping the input sequence to a value $\gamma(x) \in [-1,1]$. For closed-ended prompts, $\gamma(x) = 1$, while for open-ended prompts, $\gamma(x) = -1$.
>
> 3. Clarification of experimental details and Figure 1
>
> We have added detailed descriptions for all plots in the paper, including Figure 1. The x-axis represents the response index, sorted by reward from smallest to largest, and the y-axis represents the reward associated with each response. The solid curve indicates the mean across multiple prompts, while the shaded area represents the standard deviation. The figure clearly shows the progressive convergence of the distribution toward a single value, illustrating the reward collapse phenomenon.
>
> 4. Discussion on alignment and the new reward model
>
> We agree that examining how the newly proposed reward model affects alignment would be a valuable addition to the paper. However, due to time and resource constraints, we were unable to conduct such experiments. The alignment impact of our method could vary depending on factors such as early stopping and model size. Our focus in this paper is to investigate the reward collapse phenomenon from a theoretical perspective and validate it empirically, which we believe is of significant interest to the community. Exploring how prompt-aware optimization can improve alignment opens a promising avenue for future research.
>
> 5. Details on prompt-aware utility functions (Section 3)
>
> We have added further explanation in Section 3.2 to clarify how the proposed optimization problem is prompt-aware and how the input sequence affects the utility function.
>
> 6. Detailed descriptions of figures
>
> In response to your suggestion, we have provided detailed descriptions for all figures in the paper to ensure that the results and their implications are clear.
>
> 7. Future work on alignment
>
> While our current work focuses on the theoretical exploration of reward collapse, we recognize the importance of understanding its implications for alignment. We have identified this as a promising direction for future work.
>
> Thank you again for your valuable feedback on our paper, and we hope our responses meet your satisfaction. We believe that the improvements guided by your feedback have strengthened our work, and we would be grateful if you could consider reflecting on these enhancements in your evaluation. If there are any further clarifications needed, please do not hesitate to reach out.

---

> > ### Comment · Reviewer_Bse6 · 2024-09-12
> > **Response**
> >
> > Thank you. My concerns are mostly addressed.

---

### Review · Reviewer_GMQ5 · 2024-08-12

**Summary Of Contributions:**

This paper identifies and studies the problem of reward collapse, where regardless of the prompt, the LLM returns the same distribution of rewards. The paper provides theoretical results showing the problem of reward collapse and provides empirical results demonstrating that reward collapse can happen when training LLMs in practice. As a remedy, the paper proposes to distinguish between open-ended and and specific prompts and to learn a different reward distribution that depends on the extra context of the prompt type.

**Audience:**

Yes

**Claims And Evidence:**

No

**Requested Changes:**

In the intro it states “As demonstrated in Figure 1, our prediction of reward collapse is in agreement with the empirical results.” However, reward collapse has only been loosely defined at this point and there is no description in the intro or caption about what the axis are, what the data is, or why the plots show reward collapse. Please add more details or move the figure so things are self contained. Even section 5.1 does not include sufficient details.

A related question is why the shown distribution is not more of a step/sigmoid shape like many of the other plots. It appears most similar to the U=-1/x.

The authors talk about open-ended and closed-ended and show different functional forms for U that are better suited for these. It would be nice to discuss what other types of prompts may exist and what other types of functional forms would be well suited. This would help improve the potential generalization of the results of the paper.

More discussion is needed about the assumption of always having n completions for each prompt in the dataset.

Reward outputs are limited to [0,1] or [-1,1], but in practice they may not be within a specific range and often RLHF rewards are just a linear combination of the layer and can be any real valued number. In this unbounded case, it is unclear whether there really is a unique maximizer since (with the log sigmoid) loss values are invariant to adding a constant to the reward, thus it seems like there will be many possible solutions that could be found. It is unclear how this effects the theory. Is the unique maximizer really unique or only up to affine shifts? It also seems like in the limit $r_1 = \infty$ and $r_n=-\infty$.

The experiment in 5.1and/or 5.2 would be stronger if the reward distributions were analyzed when training on all (or more) of the data and including instances where there are not exactly the same number of responses per prompt in the dataset. It would be beneficial to see if reward collapse is less pronounced in this setting and by how much. This would strengthen the paper by showing how the theory holds up in more realistic conditions. I'm not convinced that the current experiment justifies the current claim of "realistic conditions" in the paper.

Typos:
- Discussion: uncertaintycite
- A.2: promopt type
Page 2: “collapse reward distribution” -> “collapsed …”
2.2: First, |D|M is written and then NM is written and then later |D|M is used. It would be nice to not switch notation

**Strengths And Weaknesses:**

Strengths:
+ The idea of reward collapse appears novel.
+ The theoretical results are interesting and appear sound.
+ The empirical results show the reward collapse emerges when training LLMs on real data.
+ The plots are nice and show the emergence of reward collapse and the comparison to the theoretical predictions is also nice and builds confidence in the results and analysis.

Weaknesses:
+ Doesn’t detect the prompt type but requires this as a label. This could limit the applicability of the approach since most datasets for training LLMs do not contain this type of label.
+ Reward outputs are limited to [0,1] or [-1,1], but in practice they may not be within a specific range and often RLHF rewards are just a linear combination of the layer and can be any real valued number. In this unbounded case, it is unclear whether there really is a unique maximizer since (with the log sigmoid) loss values are invariant to adding a constant to the reward, thus it seems like there will be many possible solutions that could be found. It is unclear how this effects the theory.
+ It is unclear how the prompt aware reward model is trained. How is the prompt information fed into the model? Is it just two models for different Us and one of these is selected based on the prompt label?
+ Some results are presented too early (e.g. Fig 1) and there is not enough description or background to appreciate them.
+ Most of the analysis and the experiments relies on always having n completions to a prompt. This is not guaranteed in practice and the empirical results seem a bit stretched to try and make this assumption true. For example, Experiment 5.1 seems overly constrained without much data (only 128 prompts) and throws out all data that doesn’t contain 5 responses. This begs the question of how common reward collapse is in practice.

---

> ### Author Response · Authors · 2024-08-17
> **Response to Reviewer GMQ5**
>
> Thank you for your valuable feedback on our paper. We address the problems as follows:
>
> 1. Prompt Type Detection and Applicability
>
> > Doesn’t detect the prompt type but requires this as a label.
>
> Our current prompt-aware optimization assumes the prompt type as a label. We acknowledge that this assumption may not hold in practice. We addressed this issue by a discussion in Appendix A.2 on how to obtain the prompt type. In short, this step can be done either using existing LLMs like GPT-4 or by analyzing the variability across different responses.
>
> 2. Bounded Reward Outputs
>
> > Reward outputs are limited to [0,1] or [-1,1].
>
> It is challenging to establish a reward collapse theory if the reward is not limited to [0,1]. If the reward can be arbitrarily large or small, the function $\sum_{1\le i < j \le n } U(r_i-r_j)$ may not have a finite maximum, and the maximizer may not be unique (even up to shifts). For example, if $U(z) = z$, then the maximum is infinite and can be obtained as long as $r_1 = \infty$ and $r_i > 0$. However, if we restrict $r_i$ to a bounded interval $[-M,M]$, the problem can be reduced to our case, where reward outputs are limited to [0,1]. Therefore, we believe it is reasonable to focus on limited reward outputs.
>
> 3. Prompt-Aware Reward Model Training
>
> > It is unclear how the prompt-aware reward model is trained.
>
> We apologize for the overuse of $x$ as the input sequence and the input of the utility function. We have revised the notation in the paper. Our proposed prompt-aware optimization is general, allowing any choice of utility function $U_x$ depending on $x$. In this paper, we investigate a specific utility function $U_x(z)  = z^{\gamma(x)}.$
> Here, $\gamma$ is a function mapping any input sequence to a number $\gamma(x) \in [-1,1]$. If $x$ is a closed-ended prompt, $\gamma(x) = 1$. If $x$ is an open-ended prompt, $\gamma(x) = -1$. For an open-ended prompt $x$, $\gamma(x) = -1$. For this prompt, we use the utility function $G_x(z) = z^{\gamma(x)} = z^{-1}$.
>
> 4. Timing and Description of Results
>
> > Some results are presented too early (e.g., Fig 1), and there is not enough description or background to appreciate them.
>
> We have added detailed descriptions for each plot and improved their readability to make sure readers have enough background to appreciate them.
>
>  5. Number of Completions and Empirical Results
>
> > Most of the analysis and experiments rely on always having $n$ completions to a prompt.
>
> Our theory remains valid even if each prompt has a different number of completions. In our experiment, we simplified the scenario by assuming all data contains the same number of responses. We focused on 128 prompts to ensure that the overall utility is sufficiently optimized. Our primary goal is to identify the reward collapse phenomenon from a theoretical perspective and highlight the intrinsic disadvantage of using a single utility function for all prompts. To achieve this, we designed a small-scale experiment. We agree that stronger experiments would enhance our results, and this would be an interesting direction for future work beyond the scope of this paper.
>
> 6. Improving Figure Details and Self-Containment
>
> > Please add more details or move the figure so things are self-contained. Even Section 5.1 does not include sufficient details.
>
> We have added more details to all figures and made them self-contained. Thank you for pointing this out.
>
> 7. Reward Distribution Shape and Optimization
>
> > A related question is why the shown distribution is not more of a step/sigmoid shape like many of the other plots.
>
> This is a good question. We believe there are two reasons for this. First, the overall utility may not be sufficiently optimized in the experiment, resulting in a reward distribution slightly different from its limiting distribution. Second, when the number of completions is small, the empirical reward distribution may deviate from its limiting distribution (see Figure 2 (a) and (e) when $n = 6$). In the experiment, $n$ is chosen as $5$, which is small.
>
> 8. Discussion on Other Prompt Types and Number of Completions
>
> > It would be nice to discuss what other types of prompts may exist and what other types of functional forms would be well suited.
>
> > More discussion is needed about the assumption of always having $n$ completions for each prompt in the dataset.
>
> We have added a discussion in Section 3.2 on the types of prompts and prompt-aware optimization. Our theory can be easily generalized to the case where each prompt has a different number of completions, as our theory does not rely heavily on this assumption.
>
> Thank you again for your valuable feedback on our paper, and we hope our responses meet your satisfaction. We believe that the improvements guided by your feedback have strengthened our work, and we would be grateful if you could consider reflecting on these enhancements in your evaluation. If there are any further clarifications needed, please do not hesitate to reach out.

---

### Review · Reviewer_FNuJ · 2024-09-08

**Summary Of Contributions:**

(Note: My review is much later than others because I had declined while on vacation, but the decline went unnoticed. I was requested to submit a review later on anyways.)

The paper proves that if a strongly concave pairwise reward function is used in RLHF fine-tuning, and an LLM is trained sufficiently to massively overfit the training objective, then on the training distribution the distribution of completion rewards will collapse to a distribution independent of the prompt (after removing the translation symmetry).

**Audience:**

No

**Broader Impact Concerns:**

I have no impact concerns.

**Claims And Evidence:**

No

**Requested Changes:**

I do not think the paper should be accepted even after minor changes, as it is fundamentally arguing that something bad happens under massive overfitting, then ruling out regularization as a solution without sufficient justification. Regularization is a good solution to overfitting, including early stopping!

However, conditional on the paper being accepted at this or other venues, it is important that the authors address the two main flaws above:
1. Discuss regularization more fairly
2. Either rework the theorems to apply to the most common RLHF setup or explain why they do not apply
3. Ideally, add discussion and/or experiments about whether reward collapse occurs on the test set in addition to the training set

A few smaller notes:
1. Some references are miscapitalized (“Gpt”, etc.).
2. x-axis should be $x$-axis in latex.

**Strengths And Weaknesses:**

The paper has two main flaws:

## No mention of overfitting

Fundamentally, the paper is about overfitting: throughout ML, if you massively overfit to an objective, the results are bad in many ways. Here the way in which they are bad is that the LLM will not average across different similar prompts in order to pick up distributional information about the rewards. One example of this principle is that none of the theorems in this paper apply if there are duplicate datapoints with the same prompt and set of completions, but where the completions are in different orders: the assumptions in Theorem 1 and similar would then be violated because the LLM can't answer independently for the different datapoints. A similar situation applies more generally if prompts are similar.

The paper seems unaware of this connection to overfitting: the words "overfit", "overfitting", "regularize", and "regularization" do not occur in the paper. And the headline Figure 1 is on the training set only; as far as I can tell this first experiment does not have a separate test set, or evaluate whether reward collapse would happen off the training set.

In practice, the way one solves overfitting is not a complex rewrite of the reward function itself, but rather some form of regularization. Early stopping is the most common technique, as the paper cites but discards as "somewhat arbitrary and can make it challenging to determine the stopping point". I don't think it is reasonable to discard the idea of regularization so generically.

## The theorems do not apply to cross-entropy loss

The most common $U(x)$ used in RLHF is cross-entropy loss:

$$U(x) = \log \frac{1}{1 + e^{-x}}$$

This $U(x)$ is strictly concave, but unfortunately it is not $\mu$-strongly concave for any $\mu > 0$, as it asymptotes to the line $U(x) = 0$ for $x \gg 0$. This is problematic for the theory in the paper: Theorem 1 assumes $\mu$-strong concavity and thus does not apply, but also nothing like the conclusion of Theorem 1 can hold. This is because $S(r_1, \ldots, r_n)$ no longer has a unique maximizer: it can be driven arbitrarily close to its supremum 0 by sending each $r_k \to \pm \infty$ at appropriate rates.

---

> ### Comment · Reviewer_FNuJ · 2024-09-12
> **I'd be curious for other reviewer's takes**
>
> I'd be curious for the other reviewers' takes on the "not mentioning overfitting" and strict vs. strong convexity issues.

---

> ### Author Response · Authors · 2024-09-19
> **Response to Reviewer FNuJ**
>
> We thank the reviewer for their thoughtful feedback. We address the concerns as follows:
>
> > No mention of overfitting:
>
> Our primary objective is to identify the phenomenon of reward collapse from a theoretical perspective and highlight the inherent disadvantage of using a single utility function for all prompts.
>
> While the paper does not explicitly use the term "overfitting," we have acknowledged this issue in the context of reward collapse. Specifically, above Theorem 1, we describe the phenomenon as arising when "for any reward model sufficiently optimizes the overall utility." This is, in essence, an overfitting problem. We refer the reviewer to Theorem 1 for a more formal statement.
>
> Techniques that mitigate overfitting, such as early stopping or regularization, can also be employed to address reward collapse. However, our goal is not to outperform these techniques experimentally. Rather, we aim to bring attention to this issue and advocate for more principled methods that address the root cause, potentially leading to the development of new approaches.
>
> In the revision, we have added more fair discussions on overfitting and regularization.
>
> > The theorems do not apply to cross-entropy loss:
>
> The commonly used log-sigmoid function is derived from maximum likelihood estimation of Bradley-Terry-Luce (BTL) model.
>
> The theorems do apply to logsigmoid function because on [0,1], log sigmoind is bounded, strongly concave, and increasing. In this paper, we assume the value of reward is between [0,1]. In the revison, we have made this assumption clearer to avoid confusion.
>
> As we discussed in our response to another reviewer, extending the reward collapse theory to cases where rewards are unbounded is challenging. If the reward can be arbitrarily large or small, the function $\sum_{1\le i < j \le n } U(r_i-r_j)$ may not have a finite maximum, and the maximizer may not be unique (even up to shifts). For example, if $U(z) = z$, then the maximum is infinite and can be obtained as long as $r_1 = \infty$ and $r_i > 0$. However, if we restrict $r_i$ to a bounded interval $[-M,M]$, the problem can be reduced to our case, where reward outputs are limited to [0,1]. Therefore, we believe it is reasonable to focus on limited reward outputs.
>
> In this paper, we devoted significant attention to studying the log-sigmoid function. For instance, Theorem 4 investigates the limiting distribution when U is the log-sigmoid function. Additionally, Figure 2(e) provides a numerical estimation of this limiting distribution.
>
>
>
> At last, we greatly appreciate the reviewer’s time and effort in evaluating our paper, and we hope our responses have addressed the concerns raised.

---

> > ### Comment · Reviewer_FNuJ · 2024-09-21
> > **log sigmoid**
> >
> > It's possible I'm not understanding the log sigmoid / cross-entropy distinction. Would it be possible to get an example of an RLHF paper which uses log sigmoid with domain restricted to [0, 1], so that I can see it in context? Naively, I am not sure I've seen such a paper, and indeed the restriction seems arbitrary as 1 is  not a special value in the domain of log sigmoid.

---

> > > ### Author Response · Authors · 2024-09-24
> > > **Response to log sigmoid**
> > >
> > > To clarify, our theory applies not just to log-sigmoid functions restricted to $[0, 1]$, but also to log-sigmoid functions with any bounded domain, such as $[-M, M]$. The restriction to $[0, 1]$ is not essential, and as you rightly point out, the choice of $1$ as a boundary is only for simplicity. The underlying principles extend naturally to any bounded interval.
> > >
> > > For an example of similar assumptions, we refer you to [1], where Assumption 2.1 bounds the reward model. This reflects a common practice in the literature, where reward models are typically assumed to be bounded to ensure the existence and stability of Maximum Likelihood Estimation (MLE). At a high level, since the reward model is derived from the MLE of the Bradley-Terry-Luce (BTL) model (see Eq. 2 in [2]), it is standard to assume the ‘true’ rewards are bounded within a compact set. This is a conventional assumption for MLE, which you can also find in other contexts, such as Eq. 2.1 in [3].
> > >
> > > [1] Zhu, Banghua, Michael Jordan, and Jiantao Jiao. "Principled reinforcement learning with human feedback from pairwise or k-wise comparisons." International Conference on Machine Learning. PMLR, 2023.
> > >
> > > [2] Rafailov, Rafael, et al. "Direct preference optimization: Your language model is secretly a reward model." Advances in Neural Information Processing Systems 36 (2024).
> > >
> > > [3] Gao, Chao, Yandi Shen, and Anderson Y. Zhang. "Uncertainty quantification in the Bradley–Terry–Luce model." Information and Inference: A Journal of the IMA 12.2 (2023): 1073-1140.

---

### Author Response · Authors · 2024-08-17
**Official Comment by Authors**

Thank you to all the reviewers for providing their valuable feedback. In response to the insightful suggestions from the reviewers, we uploaded a revised version of our paper. We have made the following key changes to address the feedback received:

- Notation Revision

 We have changed all instances of the notation "x" to "z" in the utility function, now consistently denoted as $U(z)$. This change was made to clarify the distinction between the input sequence and the variables used in the utility function.

- Clarification in Section 3

We have expanded Section 3 to provide a more detailed explanation of our prompt-aware optimization based on open-endedness. We also added a discussion on the types of prompts and how to choose the utility function correspondingly.

- Enhanced Plot Descriptions

We have added detailed descriptions to all plots throughout the paper. These descriptions are now more comprehensive, ensuring that the figures are self-contained and clearly explain the results and their relevance to our study.

- Typos

We corrected some typos in the paper.

At last, we want to thank all reviewers again for their time and efforts reviewing this paper.

---

### Decision · Action_Editor_tCfs · 2024-10-13

**Recommendation:** Reject

**Comment:**

Overall I think the weakness of this paper is the connection between the assumptions (on Theorem 1) and the common practice.

More experiment results will be very useful to address the questions raised by reviewer FNuJ and GMQ5, especially on the comparison between the new reward setup and the early-stop, and between the bounded and unbounded reward function domains. I think this is beyond a minor revision, so I make the decision 'reject and resubmit'.

**Audience:**

1. This paper identifies a potential problem of treating all prompts with the same reward function in RLHF for training LLM, i.e., reward collapse, which is an intriguing phenomenon.
2. This paper provides an alternative to regularization methods like early-stop to prevent reward collapse.

**Claims And Evidence:**

Strength:
1. Given the assumption made in this paper, the theoretical results are sound, and the empirical results are useful to confirm the theoretical results.

Weakness:
1. In the abstract, the authors said they introduced a 'prompt-aware optimization scheme' while the proposed scheme relies on the additional information, prompt type, as a label instead of the prompt itself, which limits the applicability.
2. There is not enough evidence on 'whether reward collapse happens in practice', especially when the common setup uses early-stop and unbounded reward function domain.

**Resubmission Of Major Revision:**

The authors may consider submitting a major revision at a later time.